# Performance of Multilayer Composite Hollow Membrane in Separation of CO_2_ from CH_4_ in Mixed Gas Conditions

**DOI:** 10.3390/polym14071480

**Published:** 2022-04-05

**Authors:** Shahidah Zakariya, Yin Fong Yeong, Norwahyu Jusoh, Lian See Tan

**Affiliations:** 1Chemical Engineering Department, Universiti Teknologi PETRONAS, Seri Iskandar 32610, Perak, Malaysia; shahidah.zakariya@utp.edu.my (S.Z.); norwahyu.jusoh@utp.edu.my (N.J.); 2CO_2_ Research Centre (CO2RES), R&D Building, Universiti Teknologi PETRONAS, Seri Iskandar 32610, Perak, Malaysia; 3Department of Chemical Process Engineering, Malaysia-Japan International Institute of Technology (MJIIT), Universiti Teknologi Malaysia (UTM), Jalan Sultan Yahya Petra, Kuala Lumpur 54100, Malaysia; tan.liansee@utm.my

**Keywords:** dip-coating, metal organic frameworks (MOFs), multilayer composite hollow fiber membrane, CO_2_/CH_4_ mixed gas separation

## Abstract

Composite membranes comprising NH_2_-MIL-125(Ti)/PEBAX coated on PDMS/PSf were prepared in this work, and their gas separation performance for high CO_2_ feed gas was investigated under various operating circumstances, such as pressure and CO_2_ concentration, in mixed gas conditions. The functional groups and morphology of the prepared membranes were characterized by Fourier transform infrared spectroscopy (FTIR) and field emission scanning electron microscopy (FESEM). CO_2_ concentration and feed gas pressure were demonstrated to have a considerable impact on the CO_2_ and CH_4_ permeance, as well as the CO_2_/CH_4_ mixed gas selectivity of the resultant membrane. As CO_2_ concentration was raised from 14.5 vol % to 70 vol %, a trade-off between permeance and selectivity was found, as CO_2_ permeance increased by 136% and CO_2_/CH_4_ selectivity reduced by 42.17%. The membrane produced in this work exhibited pressure durability up to 9 bar and adequate gas separation performance at feed gas conditions consisting of high CO_2_ content.

## 1. Introduction

Natural gas is a more desirable power source than coal since it has a lower carbon impact [1,2]. Natural gas usually contains 50% to 90% methane (CH_4_); nevertheless, harmful contaminants such as water (H_2_O), carbon dioxide (CO_2_), hydrogen sulfide (H_2_S), nitrogen (N_2_), ethane (C_2_H_6_), propane (C_3_H_8_), and toluene are often found in unprocessed natural gas [3]. With the presence of H_2_O, the acid gases CO_2_ and H2S may damage the processing and transportation equipment; thus, raw natural gas must be treated before use [4,5]. In many petroliferous basins, especially in Southeast Asia, high carbon dioxide levels in reservoirs make exploration challenging. The offshore field in Malay Basin’s reservoir usually poses high CO_2_ concentrations, making exploration difficult. Some fields contain more than 80% CO_2_, making them undesirable development prospects [6].

CO_2_ removal is crucial in the natural gas purification process. It causes corrosion in pipelines, lowers the calorific value of natural gas, and raises maintenance and operating costs [7]. For the past few decades, membrane technology has reigned supreme in gas separation processes due to its low cost and ease of processing [8,9]. However, the trade-off between permeability and selectivity limits the gas separation performance of the commercially used polymeric materials [10].

Progress in polymeric materials for gas separation has accelerated dramatically in the past few decades, including polysulfone (PSf), cellulose acetate (CA), polyethersulfone (PES), and the polyimide family [11]. PSf has been widely explored and used for membrane separation among polymeric materials due to its lower material cost and suitable mechanical strength, thermal stability, chemical stability, and gas permeation [12]. However, the well-known “trade-off” between permeability and selectivity caused by the formation of defects, such as the presence of macrovoids on the fiber surface, has resulted in poor gas selectivity [8,9].

To overcome these limitations, various techniques have been proposed, including polymer blending, ultraviolet-assisted graft polymerization, plasma-induced graft polymerization, incorporation of fillers into polymer membrane matrix, and a caulking technique that involves coating the defective membrane skin with highly permeable polymers [13,14]. Surface coating, which is typically coated on porous membrane supports with a highly permeable gutter layer and a selective layer, is one of the most effective ways to improve membrane performance in gas separation [15].

Dip-coating is a popular technique for producing thin-film composite hollow fiber membranes. Thin-film composite membranes with several layers are being developed for use in gas separation applications to enhance the efficiency of thin-film composite membranes. To cover existing flaws and protect the selective layer from abrasion or harmful chemical assaults, the protective layer is usually applied on top of the selective layer [16] On the substrate surface, the gutter layer is applied to enhance adhesion between the selected selective layer and substrate. Additionally, the gutter layer may help to reduce mass transport resistance since it is usually constructed of highly permeable materials [17].

The gutter layer acts as a bridge between the hollow fiber substrate and the selective layer, while the ultra-thin selective layer separates the gases [18]. It is typical to utilize the Polydimethylsiloxane (PDMS) coating as a gutter layer to smooth the surface, close the macrovoid, and prevent polymers from penetrating into the porous substrate [15]. However, PDMS coatings suffer from low surface energy that can cause poor interfacial adhesion between the gutter layer and the selective layer [19]. As an alternative, a composite selective layer containing inorganic fillers was incorporated into the membrane matrix to improve gas permeability [15]. Typically, rubbery-type polymers are employed because of their softness and flexibility, as well as their controlled gas penetration characteristics due to their solubility selectivity [20]. Many researchers utilize poly(ethylene oxide) (PEO) among various rubbery materials instead of PDMS since it has been identified as the preferable chemical group that interacts effectively with CO_2_ [17,18].

Polyether block amide (PEBAX) is a commercially available copolymer composed of polyamide and PEO that is well suited for use as a selective layer material. The benefits of this polymer include high skin formation ability and solvent resistance [21]. Chen et al. used the dip-coating method to prepare PEBA/PDMS/PAN multilayer composite hollow fiber membranes (HFMs)for flue gas treatment. Coating parameters such as polymer content and coating duration were studied, and they found that CO_2_ permeance of the composite membranes was improved [20].

On the other hand, over the years, many efforts have been undertaken to develop mixed matrix membranes (MMMs) for gas separation in order to overcome the limitations of polymeric materials. Lately, an MMM consisting of a new type of inorganic filler, metal organic frameworks (MOF), has been widely reported. This type of filler exhibits excellent interaction with polymers owing to its organic linkers and open metal sites [22]. One of the MOF species that shows high porosity is functionalized titanium, also known as NH_2_-MIL-125(Ti). In a recent work, Nadia Hartini et al. (2020) incorporated NH_2_-MIL-125(Ti) into a 6FDA–durene polymer matrix for CO_2_/CH_4_ separation. Membranes loaded with 7.0 wt.% of filler showed the highest CO_2_ permeability and CO_2_/CH_4_ selectivity, surpassing the 2008 Robeson upper bound [23]. Similarly, Waqas Anjum et al. found that although employing both MIL-125 and NH_2_-MIL-125(Ti) fillers enhances overall separation performance, the NH_2_-functionalized filler is recommended since it leads to better selectivity and permeability [24].

In our previous work, we investigated single gas performance of a series of composite membranes containing different compositions of NH_2_-MIL-125(Ti) in PEBAX, coated on a PSf hollow fiber support layer [25]. Enhancement of CO_2_ and CH4 gas permeance was discovered for composite membranes when compared to the PSf membranes coated only with PDMS or PEBAX solutions. Furthermore, the largest increment in CO_2_/CH_4_ ideal selectivity was found for a composite membrane loaded with 10% of NH_2_-MIL-125(Ti) filler [25]. The key reasons for the improvement in CO_2_ removal from CH_4_ are the high porosity and strong CO_2_ affinity of NH_2_-MIL-125(Ti) filler [25].

Currently, most of the research on membrane development is concentrated on single gas permeation and draws conclusions about membrane performance based on these data. This technique may cause inaccurate results owing to the lack of impurities and multicomponent gas effects, which greatly degrade pure gas performance [26].

Significant research has been conducted throughout the past few decades, with an emphasis on the modification of various polymeric precursors in the formation of hollow fiber membranes and evaluation of the resultant fibers in single gas permeation. In contrast, relatively few literature works concentrate on binary gas separation [27]. Hence, in this work, we further explore the performance of our previously developed composite hollow fiber membrane in CO_2_/CH_4_ separation in mixed gas conditions at various operating conditions such as CO_2_ feed concentration and pressure. Although in real natural gas purification processing, other impurities are present in the feed stream, the performance of the membrane in CO_2_ and CH_4_ binary gas mixture separation still could serve as the initial performance indicator prior to upscaling the membrane in real gas separation conditions [28].

## 2. Materials and Methods

### 2.1. Chemicals and Materials

Polysulfone, Mw 35,000 supplied from Sigma-Aldrich (St. Louis, MO, USA), was utilized as the polymer matrix phase for the creation of the hollow fiber membrane substrate. N,N-dimethylacetamide (DMAc), ethanol, and tetrahydrofuran (THF) were supplied by Merck and used as received. Polydimethylsiloxane (PDMS) coating layer was supplied by Sigma-Aldrich (St. Louis, MO, USA). Hexane supplied by Merck (Darmstadt, Germany) was utilized as the solvent in the preparation of PDMS coating solutions. Commercial PEBAX MH-1657 polymer was purchased from Arkema Group (Colombes, France). Previously self-synthesized NH_2_-MIL-125(Ti) particles were used as fillers [23].

### 2.2. Fabrication of PSf Hollow Fiber Substrates

The formula for preparing the dope solution is described in detail in our previous work [25]. With the dry/wet spinning process, PSf hollow fiber was spun using a spinneret with dimensions of OD/ID of 0.80 mm/0.4 mm at an air gap distance of 3.0 cm, while the take-up speed was maintained at 5.0 rpm. Then, fibers were immersed in water to remove the solvent residue for three days. Wetted fibers were then washed three times with methanol and n-hexane for 30 min each time. The solvent-exchanged fibers were then dried at room temperature before being subjected to characterization and gas permeation experiments [29].

### 2.3. Preparation of Gutter Layer and Selective Layer

The coating solution of the gutter layer was prepared by stirring 3 wt.% PDMS in n-hexane. The coating solution of the selective layer was prepared by dissolving PEBAX pellets in a 70/30 ethanol/water solvent mixture at a concentration of 2%. The mixture was agitated under reflux at 85 °C for approximately 2 h until it was fully dissolved, and then a 5–20 wt.% loading of NH_2_-MIL-125(Ti) particles synthesized in our previous work (surface area of 1205.9 m^2^ g^−1^ and pore volume of 0.53 cm^3^ g^−1^) [23] was added to the solution. Prior to coating, the NH_2_-MIL-125(Ti)/PEBAX suspension was alternately stirred and sonicated for 30 min to ensure complete dispersion of particles in the solution. Subsequently, this solution was stirred and sonicated again to remove any bubbles formed prior to coating. The hollow fiber membranes were first dip-coated for 10 min with PDMS solution as a gutter layer. Then, the coated hollow fibers were dried for 24 h before being coated with NH_2_-MIL-125(Ti)/PEBAX solution. Finally, the composite hollow fibers were cured at room temperature for 48 h before proceeding to gas separation testing. The membranes developed in our previous work [25] and used in this study are shown in Table 1.

### 2.4. Characterization of Hollow Fiber Membranes

The crystallinity of all composite membranes was examined by using an X-ray diffractometer (X’Pert3 Powder, Panalytical, Malvern, UK) with Cu Kα radiation at ambient temperature. The surface of each hollow fiber sample was irradiated with X-rays and the intensities and scattering angles of the X-rays that leave the samples were measured from 2θ values of 5° to 35°. In addition, attenuated total reflectance (ATR)-FTIR was used to acquire infrared spectra of the resulting membranes. A total of 50 scans with wavenumbers ranging from 650 to 4000 cm^−1^ were used to obtain the spectrum of the outer surface of each hollow fiber membrane with a sample size of 1 cm. The morphology of hollow fiber membranes was examined by field emission scanning electron microscopy (FESEM) using a Zeiss Supra 55VP (Jena, Germany). The membrane surface was analyzed for elemental composition using a dispersive X-ray spectrometer (EDS), Bruker Quantax 70 (Berlin, Germany), to confirm the presence of Ti in the NH_2_-MIL-125(Ti) particle in the coating layer.

### 2.5. CO_2_/CH_4_ Binary Gas Separation Testing

The module was produced by assembling a few 9 cm long fibers prior to the mixed gas permeation test, as illustrated in Figure 1. Both sides of the module were sealed using a 5 min high-performance epoxy glue that was then allowed to dry for 24 h. The module was then placed in a stainless steel pressure chamber for the gas separation test. The binary gas permeability of the resulting membrane was tested from 1 to 9 bar using CO_2_/CH_4_ binary mixtures containing 14.5 vol %, 42.5 vol %, and 70.0 vol % of CO_2_. Gas chromatography (Perkin Elmer, model GCNARL9680, Waltham, MA, USA) equipped with a thermal conductivity detector (TCD) was used to evaluate the gas compositions of feed, retentate, and permeate gas streams. The full experimental and set-up methods have been published elsewhere [30]. The permeability of each gas was calculated by using Equations (1) and (2), which are as follows [31]:(1)$$P_{{\mathrm{CO}}_{2}}=\frac{V_{p}y_{{\mathrm{CO}}_{2}}t}{A_{m}\left(P_{h}x_{{\mathrm{CO}}_{2}}-P_{l}y_{{\mathrm{CO}}_{2}}\right)}\;P_{{\mathrm{CO}}_{2}}=\frac{V_{p}y_{{\mathrm{CO}}_{2}}t}{A_{m}\left(P_{h}x_{{\mathrm{CO}}_{2}}-P_{l}y_{{\mathrm{CO}}_{2}}\right)}$$
(2)$$P_{{\mathrm{CH}}_{4}}=\frac{V_{p}y_{{\mathrm{CH}}_{4}}t}{A_{m}\left(P_{h}x_{{\mathrm{CH}}_{4}}-P_{l}y_{{\mathrm{CH}}_{4}}\right)}$$
where $P_{CO_{2}}$, $V_{p}$, $A_{m}$, $P_{h}$, $P_{l}$, ***x***, and ***y*** are CO_2_ permeability (GPU) in the gas mixture, permeate flow rate (cm^3^(STP)/s), membrane area (cm^2^), feed pressure (bar), permeate pressure (bar), and the mole fractions of the component in the feed and permeate streams, respectively. The same equations were used to determine the CH_4_ permeability in the gas mixture.

The CO_2_/CH_4_ mixed gas selectivity was calculated using Equation (3) as follows [32]:(3)∝CO2CH4=yCO2yCH4xCO2xCH4

## 3. Results and Discussion

### 3.1. X-ray Diffraction (XRD)

The X-ray diffraction patterns of the resulting membranes are shown in Figure 2. Normally, a polymer sample with an amorphous region exhibits a wide peak intensity [33]. From the results obtained, the XRD pattern for almost all hollow fiber membranes showed a wide band between 15° and 20°. By embedding the particles in the polymer matrix, the membranes became more amorphous, and this result is consistent with the previous results described by Ghasemi et al. [34]. Moreover, the membranes’ broad peaks were attributed to the compatibility and full homogeneity of membrane components [35]. Following the integration of the MOFs, the peak locations remained unchanged, demonstrating that there were no changes in the d-spacing of the polymer [27].

### 3.2. Fourier Transform Infrared Spectroscopy (FTIR)

Figure 3 shows the FTIR spectra of PSf hollow fiber membranes coated with PDMS, PEBAX, and NH_2_-MIL-125(Ti)/PEBAX containing 5, 10, 15, and 20 wt.% NH_2_-MIL-125(Ti) particles. The FTIR spectrum of NH_2_-MIL-125(Ti) shows a broad peak between 3400 and 3700 cm^−1^, ascribed to –NH_2_’stretching vibration [36]. Peaks between 1658 and 1253 cm^−1^ shown by NH_2_-MIL-125(Ti) fillers correspond to carboxylic acid functional groups within the MOF structure [37]. Meanwhile, asymmetric stretching vibration bands at 1654 cm^−1^ (C=O) and symmetric stretching vibration bands at 1253 cm^−1^ (C–O) observed in the spectrum are attributed to the presence of carbonyl groups in the filler [38]. Peaks between 500 and 800 cm^−1^ are attributed to the O-Ti-O vibration [39]. These remarkable peaks demonstrate the successful synthesis of NH_2_-MIL-125(Ti). On the other hand, a band at 793 cm^−1^ shown in the FTIR spectrum of PDMS/PSf membrane (C) corresponds to the stretching vibration of Si–O bonds. The presence of this band in the FTIR spectrum indicates the presence of PDMS sub-chains in the membrane [40]. Additionally, peaks at 2966, 1102, and 1014 cm^−1^ shown in the PDMS/PSf (C) spectrum correspond to the C–H stretching vibrations of Si-CH_3_ and Si-O-Si [41].

For the PEBAX/PDMS/PSf (C_0_) membrane, the distinct peak at around 1238 cm^−1^ is attributed to the stretching vibration of the C–O–C group within the PEO segment [42]. Furthermore, the membrane exhibits relatively sharp peaks at 3301, 1488, and 1641 cm^−1^. These peaks are attributed to the hard polyamide segment’s –N–H–, H–N–C=O, and O–C=O groups [42]. Referring to Figure 3, membranes C_5_-C_20_ exhibit minor bands from 3400 to 3700 cm^−1^, corresponding to the –NH_2_ stretching vibration from the particles. Considering this, the bands associated with the PEBAX selective layer are stronger, indicating that the PDMS bands detected might be caused by the PEBAX layer.

It can be seen from Figure 3 that membranes C_0_-C_20_ exhibit similar FTIR spectra. However, in comparison with the C_0_ membrane, the reduced peak at 1253 cm^−1^ in the FTIR spectrum of the membrane C_10_ indicates the interaction of PEBAX and NH_2_-MIL-125(Ti). This observation shows that the NH_2_-MIL-125(Ti) particles on the surface of fibers disturbed the chain of PEBAX. Additionally, no new peaks were found in the FTIR spectra of composite membranes (C_5_-C_20_), indicating that the NH_2_-MIL-125(Ti) and PEBAX were physically blended [43].

### 3.3. Field Emission Scanning Electron Microscopy (FESEM)

Figure 4 shows the FESEM images of PSf/PDMS membrane (C) at the outer surface. The outer surface of the membrane after PDMS coating became denser and smoother. This may help eliminate the solution intrusion during the subsequent coating of the selective layer containing NH_2_-MIL-125(Ti) in PEBAX.

The outer skin, which is composed of a PEBAX/NH_2_-MIL-125(Ti) selective layer at various filler loadings, is responsible for gas separation, whereas the porous sublayer beneath offers both mechanical support and separation [9]. FESEM images of the PSF hollow fiber coated with PDMS as the first layer and NH_2_-MIL-125(Ti)/PEBAX as a subsequent layer are shown in Figure 5. From Figure 5, it can be seen that a modest particle dispersion with the same thickness was found for all membranes (Figure 5a–c), where the concave surface is visible and smaller particles were most likely present in the coating dispersion, leading them to adhere to the membrane surface. This is owing to the flexibility of the PEBAX chains, which enables superior contact and adhesion with the NH_2_-MIL-125(Ti) particle [44]. Figure 5d shows a slight reduction in the thickness of the membrane loaded with 20% NH_2_-MIL-125(Ti). Ultimately, all the images demonstrate that the PEBAX coating layer provides a conducive environment for the adhesion of NH_2_-MIL-125(Ti) particles to the membrane surface. In an earlier work, our EDX mapping analysis was performed on the membrane surface to determine the distribution of NH_2_-MIL-125(Ti) particles in the outer coating layer [25]. The existence of NH_2_-MIL-125(Ti) on the membrane surface was confirmed by scanning the elements of titanium, the major component of NH_2_-MIL-125(Ti). Certainly, the dispersion of titanium increased with higher particle loadings.

### 3.4. CO_2_/CH_4_ Mixed Gas Separation Performance

#### 3.4.1. Effect of CO_2_ Concentration in Feed Stream

Our previous study found that the best single gas permeation performance was exhibited by a composite membrane loaded with 10 wt.% filler (C_10_) [25]. In the present work, we further explore the performance of this membrane in CO_2_/CH_4_ separation in mixed gas conditions. Figure 6 shows the effect of CO_2_ feed composition on CO_2_ and CH_4_ permeances as well as the selectivity in mixed gas separation evaluated at 25 °C for the C_10_ membrane. The CO_2_ concentrations ranged from 14.5 vol % to 70 vol % at a feed pressure of 5 bar. CO_2_ is well known as a plasticizer for polymeric membranes. The higher the CO_2_ content in the membrane, the greater the polymer free volume and segmental mobility, resulting in a decrease in membrane selectivity [45].

As seen in Figure 6, CO_2_ permeance steadily increases as CO_2_ concentration increases, and vice versa for membrane selectivity. At a CO_2_ feed concentration of 70 vol %, a maximum CO_2_ permeance of 15.10 GPU is attained. Meanwhile, with a CO_2_ feed concentration of 14.5 vol %, a minimum CO_2_ permeance of 6.4 GPU is attained. Furthermore, under equal operating circumstances, the CO_2_ permeance increase is modest for CO_2_ concentrations below 40 vol %, being around 35%, compared to that for CO_2_ concentrations beyond 40 vol %, which is about 76%.

However, the results demonstrate that selectivity declined as CO_2_ feed concentration increased. The CO_2_/CH_4_ mixed gas selectivity showed a substantial decline from CO_2_ feed concentrations of 14.5 vol % CO_2_ to 70 vol % CO_2_ (about 42%). Lower CO_2_/CH_4_ mixed gas selectivity was observed at higher CO_2_ concentrations, despite the membrane showing larger CO_2_ adsorption potential. This phenomenon is mainly due to the greater CH_4_ adsorption capability of the membrane, which reduced the mixed gas selectivity [44]. As a result, a maximum CO_2_/CH_4_ mixed gas selectivity of 7.9 was obtained at a CO_2_ feed concentration of 14.5 vol %. Furthermore, increasing the CO_2_ feed concentration from 42.5 vol % to 70 vol % resulted in the saturation of the amine–CO_2_ interaction, which aggregated CO_2_ on the feed side of the membrane, thus lowering the CO_2_/CH_4_ mixed gas selectivity [46]. Additionally, a larger CO_2_ feed concentration might inflate the polymer matrix, resulting in an increase in the rate of CH_4_ penetration through the membrane [47].

From the results obtained in this work, we found that the selectivity of mixed gas is less than that of pure gases [48]. However, the mixed gas selectivity of CO_2_/CH_4_ is greater than the CO_2_/CH_4_ ideal selectivity at a CO_2_ feed concentration of 14.5%, indicating that CO_2_ and CH_4_ compete for the adsorption site in the membrane. In comparison, at a CO_2_ feed concentration of 70 vol %, the CO_2_ permeance rose by 112%, up to 15.1 GPU, compared to 7.1 GPU for pure gas permeation. Moreover, CO_2_/CH_4_ mixed gas selectivity reduced from 7.9 (CO_2_ feed concentration of 14.5 vol %) to 4.6 (CO_2_ feed concentration of 70 vol %), which is less than the CO_2_/CH_4_ ideal selectivity of 11.9 obtained in our previous work [25].

#### 3.4.2. Effect of Feed Pressure

We further conducted the separation experiment on the C_10_ membrane at different pressures up to 9 bar, and Figure 7 illustrates the effect of feed pressure from 1 to 9 bar on the performance of the C_10_ membranes at 42.5 vol % CO_2_ feed concentration.

Referring to Figure 7, increasing the feed pressure caused the increment of CO_2_ and CH_4_ permeance, as well as CO_2_/CH_4_ mixed gas selectivity. The maximum CO_2_ permeance of 11.8 GPU was obtained at 9 bar. Meanwhile, at 1 bar, a minimum CO_2_ permeance of 5.0 GPU was achieved. The CO_2_ permeability rose 135%, from 5.0 GPU to 11.8 GPU, when the pressure was raised from 1 to 9 bar. However, a distinct pattern can be seen for the CH_4_ permeability. It remained relatively consistent between 1.5 GPU and 1.8 GPU when the pressure increased from 1 to 9 bar. This phenomenon could be explained by greater CO_2_ condensability as a result of its increased sorption capability.

Moreover, the increment of CO_2_ permeance at higher pressures could be also related to the increase in gas solubility, caused by the enhancement of CO_2_ molecule sorption in the polymeric network, where the CO_2_ fills the gap between the polymer network’s chains. This widens the distance between these bonds, and thus increases the mobility of the polymeric chain [46] and plasticizes the membrane. Eventually, the gas permeance and the gas compressibility of the membrane increase [49]. For all pressures investigated in this experiment, CO_2_ permeance rose roughly linearly with increasing pressure, but CH_4_ permeability decreased, showing competition for adsorption sites and, once again, preferential adsorption of CO_2_ over CH_4_ [50].

Furthermore, by increasing the feed pressure, CO_2_/CH_4_ mixed gas selectivity was also increased. When the pressure increased from 1 to 9 bar, the selectivity increased from 2.9 to 7.2 (Figure 7). This result is mainly due to higher CO_2_ condensability compared to CH_4_ (Tc of CO_2_ is 31.1 °C compared to 82.3 °C for CH_4_), which resulted in a stronger affinity of CO_2_ to the membrane. Moreover, the kinetic diameter of CO_2_ of 3.3 Å is smaller than that of CH_4_ (3.82 Å); therefore, the penetration rate of CO_2_ over the membrane was greater than CH_4_ [51]. In addition, the increase in mixed gas selectivity is also due to the inherent flexibility of NH_2_-MIL-125(Ti) filler.

As can be seen from Figure 7, the CO_2_ permeance and CO_2_/CH_4_ mixed gas selectivity of the membrane increase with increasing feed pressure. These results reveal that satisfactory separation performance can be maintained at higher pressure. Thus, it can be deduced that that the PSF/PDMS/PEBAX/NH_2_-MIL-125(Ti) membrane prepared in this work can be considered as a promising candidate for practical membrane-based natural gas purification.

## 4. Conclusions

Multilayer composite hollow fiber membranes containing NH_2_-MIL-125(Ti) particles were fabricated using the dip-coating technique and assessed for CO_2_/CH_4_ separation at various CO_2_ feed concentrations and feed pressures. Additionally, the chemical structure, phase structure, and morphology of the membrane were studied using different analytical tools. The XRD patterns showed the typical NH_2_-MIL-125(Ti) structure peaks with an amorphous state in the membranes, and no crystallization of the NH_2_-MIL-125(Ti) was found during the coating procedure in the composite membranes. FTIR results revealed that the addition of more particles into the polymer matrix resulted in no new peaks for all the composite membranes, implying the physical blending feature of NH_2_-MIL-125 (Ti) and within the PEBAX bulk. CO_2_ permeance was greatest at a 70 vol % CO_2_ feed composition, but it decreased slightly compared to single gas permeation. The highest CO_2_/CH_4_ mixed gas selectivity obtained was 7.9 at a CO_2_ concentration of 14.5 vol % and testing pressure of 5 bar. The results of the mixed gas separation analysis indicate that the fabricated composite membrane can be considered as a viable alternative membrane material for gas separation processes.

## Figures and Tables

**Figure 1 polymers-14-01480-f001:**
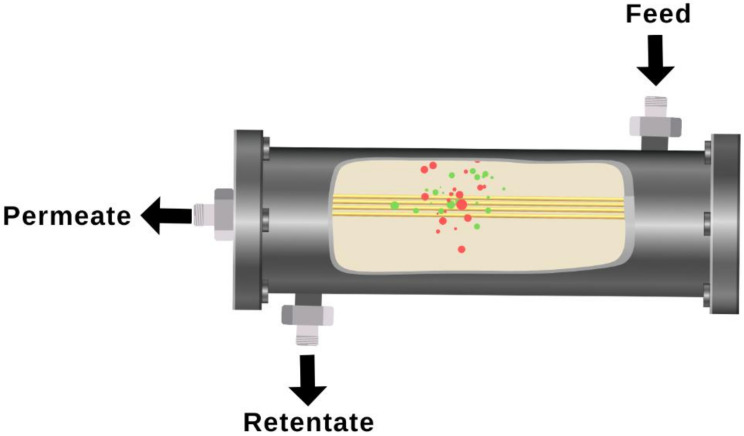
Schematic diagram of the gas permeation test module.

**Figure 2 polymers-14-01480-f002:**
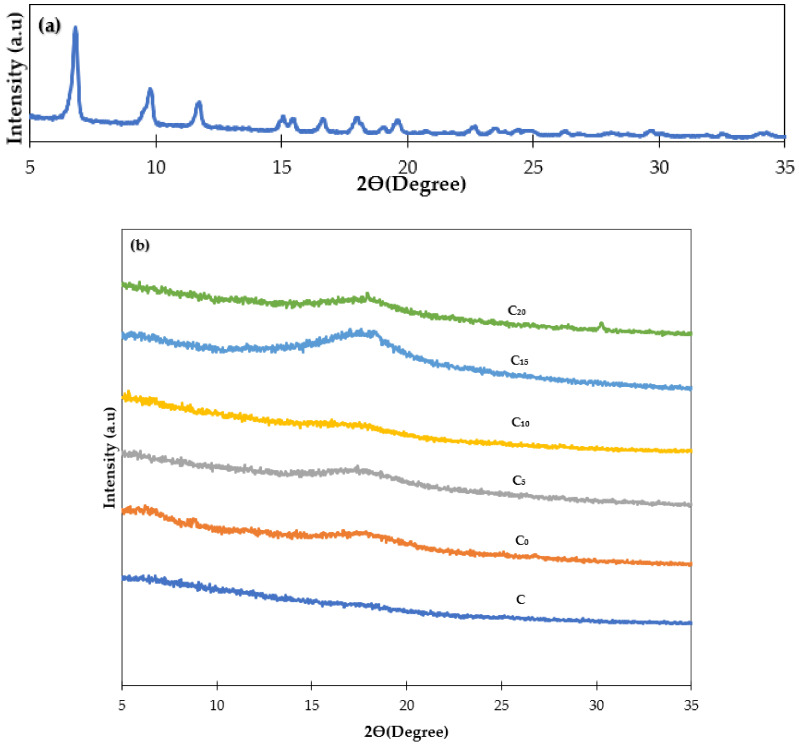
XRD patterns of (**a**) NH_2_-MIL-125(Ti) and (**b**) composite membranes.

**Figure 3 polymers-14-01480-f003:**
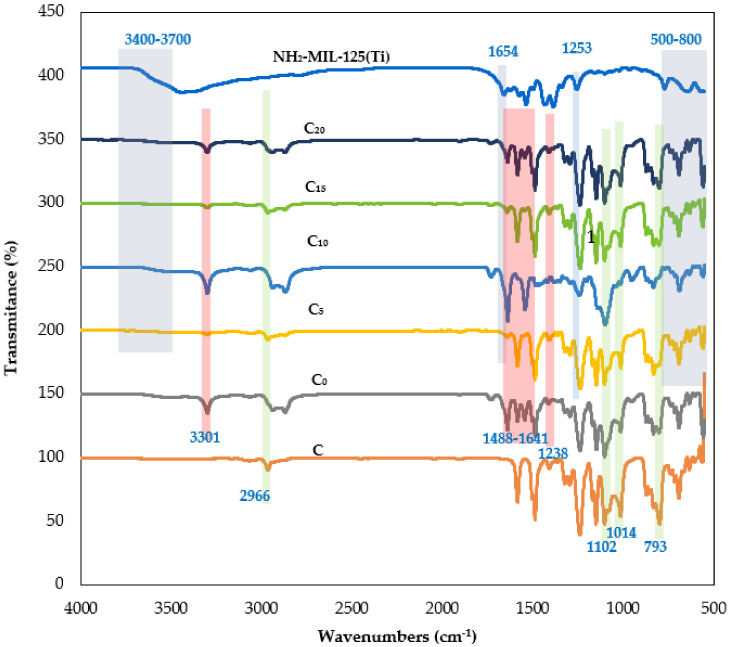
FTIR spectra of fillers and composite hollow fiber membranes.

**Figure 4 polymers-14-01480-f004:**
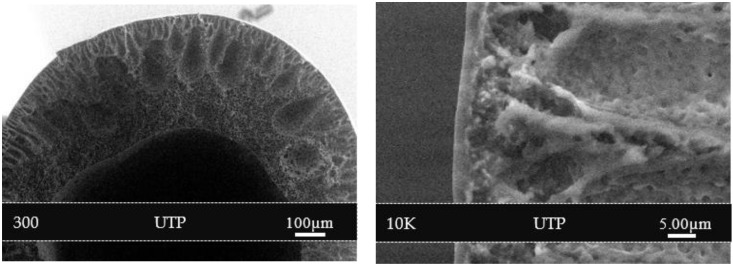
Cross-section morphology of PSf/PDMS membrane (C) at 300 and 10 K magnifications.

**Figure 5 polymers-14-01480-f005:**
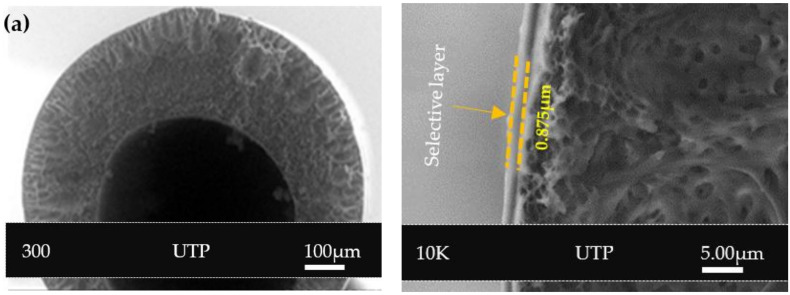
Cross-section morphology of composite hollow fiber membranes at 300 and 10 K magnifications (**a**) C_5_, (**b**) C_10_, (**c**) C_15_, and (**d**) C_20_.

**Figure 6 polymers-14-01480-f006:**
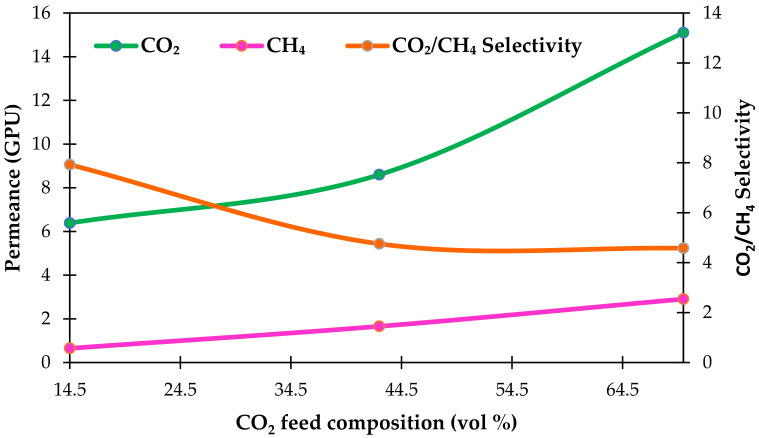
Effect of CO_2_ feed concentration on CO_2_ and CH_4_ permeances and CO_2_/CH_4_ mixed gas selectivity in mixed gas separation at feed pressure of 5 bar.

**Figure 7 polymers-14-01480-f007:**
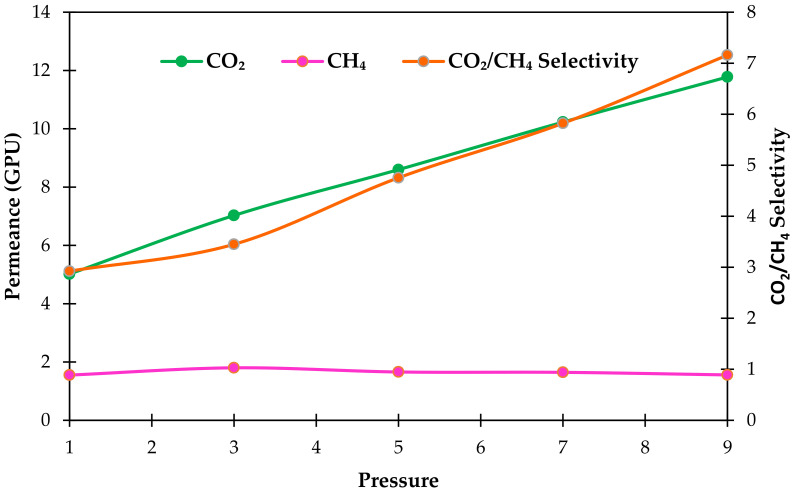
Effect of pressure on CO_2_ and CH_4_ permeances and CO_2_/CH_4_ mixed gas selectivity in mixed gas separation at 42.5 vol % CO_2_ feed concentration.

**Table 1 polymers-14-01480-t001:** Membranes prepared in our previous work [25] used in this study.

Code	Multiplayer Composite Membranes	Filler Loading (%)
C	PSf/PDMS	0
C_0_	PSf/PDMS/PEBAX	0
C_5_	PSf/PDMS/PEBAX-NH_2_-MIL-125(Ti)-5wt.%	5
C_10_	PSf/PDMS/PEBAX-NH_2_-MIL-125(Ti)-10wt.%	10
C_15_	PSf/PDMS/PEBAX-NH_2_-MIL-125(Ti)-15wt.%	15
C_20_	PSf/PDMS/PEBAX-NH_2_-MIL-125(Ti)-20wt.%	20

## Data Availability

Not applicable.

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
