# Peer review of "Performance of Multilayer Composite Hollow Membrane in Separation of CO2 from CH4 in Mixed Gas Conditions"

_polymers, 2022, doi:10.3390/polym14071480_

Round 1

Reviewer 1 Report

Re: polymers-1614027

The manuscript could be accepted for publication in Polymers subjected to major changes. The authors need to respond to the attached comments.

1) The general format of the journal was not fully adapted in many sections (e.g., the references section).

2) The abstract should concentrate on the main findings of the study. Try to avoid speculation about future research. Therefore, the abstract needs modifications. For example, delete the last sentence within the abstract section.

3) Many chemical formulas were used without giving the fall names first. They are standard chemicals but it is better to clarify for a broader readership. For example, H2O, CO2, H2S, N2, C2H6, and C3H8.

4) The introduction section is very long and should be abbreviated to be sharp and easy to follow.

5) For Figures 5 and 6, it is better to include the key within the graph not outside.

6) The conclusion section needs to be abbreviated and should be different than the abstract.

7) Correct the style of the chemical formulas in references 4, 8, 16, 30, 33, 38, 40, and others, for example.

8) Update and complete reference 11.

Reviewer 2 Report

The authors have described performance of Multilayer Composite Hollow Membrane in Separation of CO2 from CH4 in Mixed Gases Condition. Although many articles have published on the same topic including the same group of authors but only with single gas components. This manuscript deals with real mixtures od CO2 and CH4 which may find potential application in practice. The membrane is well characterized by XRD, FTIR and FESEM. The obtained results in selectivity shows how gas mixture behave differently than pure gases itself.

Author Response

N/A

Reviewer 3 Report

This manuscript reported a study upon preparing several kinds of composite membranes comprising NH2-MIL-125(Ti)/PEBAX coated on PDMS/PSf AND charactering their gas separation performance for high CO2 feed gas. The research is help to improve the treatment efficient of membrane. However, there are some issues (listed below) and should be revised and answered.

  1. For introduction, it should be accurately elucidated to show the background of current research. Thus, the latter of this section need more detailed illustration about previous researches and their existing problems, while the former of introduction should be shortly.
  2. No chemical linkage between NH2-MIL-125(Ti) and PEBAX bluk, is the nanoparticle easier to be removed? Why no peak at the band of 3400-3700 after blending from FTIR spectrum? Is it any physical connection between them?
  3. There are many porous structure existing in the membrane, it is better to quantitatively characterize the specific surface area and pore size distribution as well as pore diameter.
  4. After separation, how to deal with the CO2 in membranes so as to reuse the membranes.
  5. Page 4, line 11 is Table 1, not Table 2. Figure 2, please point out which one is a and which one is b, and in the second one of Figure 2, what does different color lines means? It is not complete for Figure 3. Some notes in Figure 6 and Figure 7 were overlapped, please modify.

Reviewer 4 Report

Please find the suggestions as the attached file.

Author Response

Thank you for the constructive comments.

Round 2

Reviewer 1 Report

Apart from the format, the manuscript is accepted for publication in Polymers
